# How weather affects cognitive and physical outcomes in older adults

Jason Shourick[1,2,3]*, Valérie Lauwers-Cances[2], Bruno Vellas[1,3,4], Nicola Coley[1,2,3], Sandrine Andrieu[1,2,3] and for the MAPT/IHU HealthAge Open Science study group[¶]

1 AGING Team, CERPOP, UMR1295, Toulouse University III, Toulouse Cedex, 2 Department of Clinical Epidemiology and Public Health, Toulouse University Hospital, Toulouse Cedex, France, 3 IHU HealthAge, Toulouse, France, 4 Gerontopole, W.H.O Collaborative Centre for Frailty, Clinical Research and Geriatric Training, Toulouse University Hospital, Toulouse, France

¶ members are listed in the acknowledgements.
* jason.shourick@univ-tlse3.fr

## Abstract

### Objective

To ascertain whether, in comparison to the participants' expected abilities, the weather may cause abnormally poor cognitive or physical performance. Design Secondary analysis of a randomised controlled trial

### Setting

Study conducted between May, 2008, and Feb, 2011 in 13 memory centres in France and Monaco

### Participants

1313 participants from the MAPT trial, a 5-year multicentre prevention trial, which included dementia-free individuals aged over 70 years. Participants presented subjective memory complaints, slow gait speed and/or an instrumental activity of daily living limitation.

### Main outcome measures

Cognition was assessed using a composite cognitive Z-score (composed of digit symbol substitution test, free and cued selective reminding test, Mini-mental state, category fluency) and subjective memory complaints. Physical function was assessed using gait speed, the short physical performance battery (SPPB) and its components, and grip strength. Abnormally low scores were defined as an observed score that was lower than the individual's expected ability by at least the minimal clinically important difference.

**Data availability statement:** Although Nicola Coley is a member of the MAPT/IHU HealthAge Open Science group, the authors do not have individual permission to share the data. The datasets generated and/or analyzed during this study are not publicly available, and have to be requested from the MAPT/IHU HealthAge Open Science group. However, individual de-identified participant data are available with immediate effect to academic researchers, following approval of a methodologically sound research proposal by the MAPT data sharing committee and signature of a data use agreement. Enquiries or proposals should be directed to nicola.coley@inserm.fr and guyonnet.s@chu-toulouse.fr.

**Funding:** The author(s) received no specific funding for this work.

**Competing interests:** The authors have declared that no competing interests exist.

## Results

Higher outdoor temperature was associated with a significantly increased risk of abnormally low gait speed or SPPB (respectively OR 1.13 95% CI (1.04, 1.22) and OR 1.15 95% CI (1.03, 1.29) for 10 degrees Celsius), but did not significantly increase the risk of abnormally low cognitive function.

## Conclusion

Our results suggest that weather conditions should be strongly considered when assessing the physical performance of older adults in the context of clinical practice and clinical research as examination in hot weather might lead to false conclusions on the participants' abilities.

## Introduction

The concepts of healthy aging and intrinsic capacity, which place more emphasis on function than diseases, have proven useful in forecasting the health, mortality, and quality of life of older adults in the setting of a rapidly aging population [1,2]. However, due to conflicting results from large prevention trials, clinical recommendations to maintain or slow the decline of these functions are still lacking. Targeting function, rather than disease, comes with its own methodological challenges. Due to the heterogeneity in status and trajectories, and of the modest expected effect of interventions, prevention trials require long follow-up periods and large populations [3]. Reducing heterogeneity due to testing conditions could lead to shorter, smaller, and more successful trials.

However, some sources of heterogeneity, such as the weather, are rarely taken into account. Yet, standardizing temperature during cognitive and physical evaluations or avoiding the warmest days or coldest days could be needed.

The effect of weather on physical and cognitive function is less clear, because most studies are physiological studies evaluating the effect of temperature and humidity rather than weather itself. Indeed, the effect of the outside temperature during a whole day might have a different impact on performance than the indoor temperature during a short evaluation session, conducted most of the time in a temperate air-conditioned or heated (as appropriate) environment. Outside temperature can induce dehydration and tiredness or impact mood in a way that may have a lasting effect on the participant's cognitive or physical performance. Furthermore, most previous work consisted of physiological studies of small groups of healthy young adults in highly controlled experiments, rather than everyday life [4–6]. However, weather, i.e., the state of the atmosphere at a given time and place affects individuals throughout the day and in more complex ways than temperature during evaluation. The few studies that have assessed the impact of weather (rather than temperature) on cognition and physical performance of older adults are conflicting [7–12]. Furthermore, these studies did not try to determine whether the performance of an individual could be unreflective of their actual function due to the weather at the time it was evaluated.

For the purposes of standardisation in clinical trials, or to provide accurate assessment of a participant's function, this question is of prime importance. Therefore, re-using data from a large multicentre dementia prevention trial, the objective of this study was to analyse if the weather on the day of evaluation could explain abnormally low cognitive or physical performance.

## Methods

### Population

MAPT (NCT01513252) was a 3-year, multicentre, randomised, placebo-controlled, 4-parallel groups prevention trial which enrolled 1680 participants between May, 2008, and Feb, 2011 in 13 memory centres in France and Monaco [13]. Participants were included if they were aged 70 years or older, community-dwelling, free of dementia, with either a subjective memory complaint, a limitation in one instrumental activity of daily living and/or slow walking speed. Participants were excluded if they presented confirmed dementia, a Mini-Mental State Examination (MMSe) score <24, or criteria for major depression or generalized anxiety disorder. These criteria were chosen to identify people who were believed to be at a higher risk of cognitive decline or dementia and who were in the early stages of cognitive and/or physical frailty [14].

Participants were randomized (1: 1: 1: 1) to either:

- A multidomain intervention (cognitive training, physical activity, nutritional guidance, and an annual preventive consultation)

- Omega-3 polyunsaturated fatty acids

- Both interventions (multidomain and Omega 3), or

- Placebo.

There was no significant difference in 3-year cognitive decline between any of the three intervention groups and the placebo group. MAPT was followed by MAPT-plus, a 2-year observational follow-up study [13]. The study population has been described in detail elsewhere [15]. Briefly, 42% of participants had a mild level of cognitive impairment without dementia (i.e., CDR score equal to 0.5).,42.1% of participants were considered as prefrail and 3.2% were considered as frail according to Fried's criteria [13,15].

MAPT participants' exact addresses were not recorded. Therefore, in the present analysis, we included participants followed-up in centres ≤10 km from a local meteorological station, and excluded 6 centres (334 participants) which were situated further away from a meteorological station.

### Ethics approval

The MAPT study was approved by the French Ethical Committee located in Toulouse (CPP SOOM II) and was authorized by the French Health Authority. Patients signed a written informed consent.

This study was performed after approval from the data sharing committee of the MAPT, and pseudomised data was accessed on the 7th of July 2020.

**Cognitive, physical and functional outcomes.** All outcomes were assessed during study visits at baseline and 6, 12, 24 and 36, 48 and 60 months.

For cognition, we used the MAPT trial's primary outcome, the average of Z-scores of 4 cognitive tests: free and total recall of the Free and Cued Selective Reminding test (FCSRT, 0–96 points) [16]; ten MMSe orientation items (1–10 points) [17]; Digit Symbol Substitution Test (DSST, 1–93 points) [18]; 2-minute Category Fluency [19]. As in the MAPT trial, the Z score was calculated using baseline means and standard deviation. Each component of the composite score

was also analysed separately (without normalization, i.e., Z score transformation, as this is the way they would be used in clinical practice) as well as the complete MMSe (1–30 points) and subjective memory complaints (SMC) assessed using a visual analogue scale, rated from 1–100, asking "How well does your memory work?". To facilitate readability, the composite Z-score is presented as a percentage (i.e., multiplied by 100).

As functional and physical outcomes, we used the Short Physical Performance Battery (SPPB) [20] and its subscales (gait speed (m/s, measured over 4m), chair rises (duration in seconds), balance (score ranging from 0 to 4)), the Alzheimer's Disease Cooperative Study Activities of Daily Living Scale Prevention Intervention (ADCS-ADL-PI) [21] and hand-grip strength (kg). The SPPB is a global measure of physical capacity, and mobility which comprises three evaluations (Gait Speed, Time to rise from chair and balance) each scored on a 4-point scale. The ADCS-ADL-PI is a 15 item questionnaire developed to assess early alterations of activities of daily living (like preparing meals or getting dressed). Hand-grip strength (of the predominant hand) was measured using a hydraulic hand dynamometer (Jamar® Hydraulic Hand Dynamometer; Sammons Preston, Bolingbrook, IL, USA) with a standardised procedure. After three consecutive measurements, the highest measurement was selected [22,23]. For both scores (SPPB and ADCS-ADL-PI), better function is indicated by higher values.

**Explanatory variables.** Meteorological variables were obtained through Meteo France, the French national meteorology institution. Ground observed data, temperature and humidity, measured every 3 hours, were used. As humidity also plays a role in the discomfort induced by temperature, the humidex index, which reflects the perceived temperature, was used to integrate bot humidity and temperature [24].

We used minimum, maximum and mean temperature per day and their corresponding humidex. Previous studies showed that weather may have a delayed effect on cognition [8], so sensitivity analyses were performed using the same indicators during the 3 or 10 previous days. All effects are presented for a 10-degree increase.

Season was defined as the meteorological season for the northern hemisphere. We used autumn as the reference season because it's neither the hottest nor the coldest season and because a study by Lim et al suggested that spring (as well as winter) could be associated with lower cognitive performance [25]).

## Statistical analysis

Descriptive analyses were performed using median, quartile 1 (Q1), quartile 3 (Q3) and range for continuous variables, and count and percentage for categorical variables. We also compared weather parameters in each arm using ANOVAs.

To estimate trajectories of cognitive and physical functions, linear mixed models were built for each outcome. Follow-up time (from baseline) was used as a continuous variable. Initial models included: participant-specific random slopes and intercepts, the fixed effects of time modelled time as a cubic polynomial, age and its interaction with time polynomials, sex, level of education and centre. All variables with significant effects ($p < 0.05$) were retained.

We explored the effects of weather on cognitive and physical function by integrating each centre and assessment date-specific weather variable separately into the model as explanatory variables.

To define abnormally low scores, we used conditional residuals (i.e., the difference between observed and expected values, taking into account both fixed and random effects) from the initial models (excluding weather/seasonal variables). Conditional residuals represent the distance between an observation and its expected value for a given participant at a given visit, and might indicate an abnormally low performance on that day. MAPT was originally powered to detect a difference equivalent to 30% of the FCSRT's baseline standard deviation. Therefore, we kept this threshold (corresponding to 1.21 points) to define an abnormally low FCSRT score [14]. For the other outcomes, thresholds were based on currently-accepted 1-year minimal important clinical difference (MCs [26–30]). Indeed, the MCID represents clinical meaningful change, which could therefore lead to clinical consequences (decision not to treat). MCIDs were as follows: cognitive composite Z-score: 30%SD; MMSe total score: 3 points; DSST: 6 points; gait speed: 0.05m/s; SPPB score: 0.55 points; chair stands: 5.1 seconds; grip strength: 6.5 kg. For the other outcomes, there are, to our knowledge, no reported

MCIDs. Therefore, we calculated the minimum detectable change (MDC) using the standard error of measurement (SEM) as follows: MDC = 1.96*SEM [31].

To test if the weather could explain abnormally low score, bivariate logistic regressions, using abnormally low scores (yes/no) as the outcome and each centre- and date-specific weather parameter as explanatory variables, were used.

Because previous studies had suggested a U-shaped weather effect, the presence of a second-degree effect (temperature²) was tested and retained if significant.

Because residuals are independent from explanatory variables, we did not include age, sex or education level in the abnormally low score analysis.

Sensitivity analyses were performed using alternative abnormally low scores definitions (outlined in supplementary S7 and S13 Tables).

Because linear mixed models account for missing visits, and because any missing meteorological data would likely be missing completely at random, all analyses were conducted on all available data, without imputation. Participants with missing information (on education) were excluded from the analysis.

All statistical significance thresholds were set at 0.05. Due to the exploratory nature of this analysis, we did not adjust for multiple tests. All analyses were performed in R version 4.1.3.

## Results

We analysed 1313 MAPT participants, totalling 6900 visits on 1508 different days over 7 years and 11 months (Fig 1). Each participant had between 1 and 7 visits (Median 6, Q1-Q3 [4–7]) and for each centre there were 1–8 visits on each day with at least 1 visit (Median 2, Q1-Q3 [1–3]). Participants' baseline characteristics and cognitive and physical function are described in Table 1.

The weather on study visit days is presented in Table 2. The mean temperature ranged from −9.8 to 29.3°C (median 11.9 Q1-Q3 [7.16–17.2]). The lowest temperatures were measured in Dijon on 7 February 2012 (minimal −12.9, mean −9.8, maximal −6.85) and the hottest day was in Toulouse on 22 August 2011 (minimal temperature 23.15°, mean 29 and maximal 36.9°C). The temperature was above 30°C in at least one centre on 75 days, and above 35°C on 5 days.

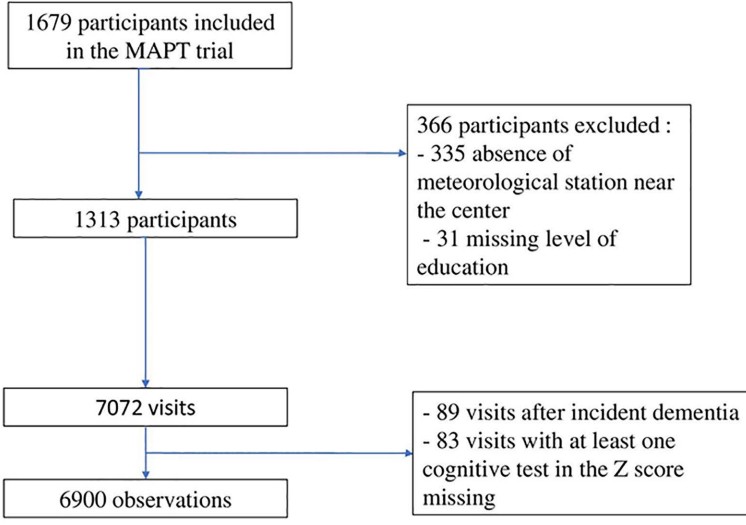

**Fig 1. Flow Chart.**

**Table 1. Description of participants at baseline.**

| | MAPT study (n = 1313) N (Percentage) or median (interquartile range) |
|---|---|
| Age, years | 75.0 [72.0-78.0] |
| Male Sex | 461 (34.2%) |
| Fried frailty phenotype | |
| Robust | 709 (55.4%) |
| Pre-frail | 531 (41.5%) |
| Frail | 39 (3.1%) |
| Primary school certificate or higher diploma | 1288 (95.5%) |
| Category Fluency* | 25.0 [20.0-31.0] |
| Mini-Mental State Examination (Orientation, range 0–10) | 10.0 [10.0-10.0] |
| Mini-Mental State Examination (Total,range 0–30) | 28.0 [27.0;29.0] |
| Free and Cued Selective Reminding test (range 0–96) * | 74.0 [68.0-79.0] |
| Digit Symbol Substitution Test (range 1–93) * | 38.0 [31.0-45.0] |
| Subjective memory performance (rang 0–100) * | 50.0 [40.0-62.0] |
| Gait Speed (meter/second)* | 1.0 [0.9-1.2] |
| Alzheimer's Disease Cooperative Study Activities of Daily Living Scale Prevention Intervention (ADCS-ADL-PI, range 0–45) * | 41.0 [37.0-43.0] |
| Short Physical Performance Battery (SPPB) (0–12) * | 11.0 [10.0-12.0] |
| SPPB Subscale: Time to rise from chair ** (seconds) | 10.7 [9.1;13.1] |
| SPPB Subscale: Balance (range 0–4) * | 4.0 [3.0-4.0] |
| SPPB Subscale: Grip Strength (kilogram) * | 26.0 [21.0-34.0] |

**Table 2. Description of the weather on visit days.**

| Total | 3448 Centre-Days* |
|---|---|
| Season (Count) | |
| Autumn | 884 (25.6%) |
| Spring | 630 (18.3%) |
| Summer | 921 (26.7%) |
| Winter | 1013 (29.4%) |
| Temperature °C, Median [IQR] | |
| Minimum | 8.35 [3.65–13.1] Min-Max [−12.9–25.1**] |
| Mean | 11.9 [7.16–17.2] Min-Max [−9.84–29.3] |
| Maximum | 15.9 [10.8–22] Min-Max [−6.85–36.9] |
| Humidex, Median [IQR] | |
| Minimum | 8.08 [1.92–14.6] Min-Max [−17.3–32.4] |
| Mean | 12.1 [5.83–19.3] Min-Max [−14.2–35.6] |
| Maximum | 16.4 [9.96–24] Min-Max [−11.2–40.7] |

*One centre-day means that the unit of description is a day in a centre where at least one visit took place on that day in this centre during the study.

** Of note the maximum daily minimum temperature didn't happen on the hottest day described in the text.

IQR: Quartile 1–3, Min: Minimum, Max: Maximum.

There was no significant difference in weather or seasonality parameters between treatment groups (Supplementary S1 Table).

## Effect on cognitive outcomes

At least one seasonal effect was observed for most of the cognitive assessments, except for the DSST and total MMSE. In winter, compared to autumn, the composite Z-score increased by 4.87% and category fluency increased by 0.582 points, whereas subjective memory scores decreased by 1.37 points. Compared to autumn, the FCSRT decreased by 0.639 points in summer (Supplementary S2 Table).

Temperatures or humidex only affected the FCSRT and subjective memory performance. A 10-degree increase in minimum, mean or maximum temperature was associated, respectively, with decreases of 0.381, 0.346 or 0.26, in the FCSRT, and increases of 0.683, 0.522 and 0.322 (not significant) in subjective memory performance (Supplementary S2 Table).

Sensitivity analyses using three- or ten-day lags were consistent with the main analysis, showing detrimental effects of increased temperature and humidex on the FCSRT and positive effects on subjective memory (Supplementary tables 3 and 4).

## Abnormally low cognitive performance analysis

Among the 6900 visits, the number of instances of abnormally low scores was 812 (11.8%) for the composite Z score, 269 (3.90%) for the DSST, 535 (7.75%) for category fluency, 1260 (18.26%) for the FCSRT, 80 (1.16%) for the MMSe, and 108 (1.57%) for subjective memory performance.

Regarding seasonality, compared to autumn, there was a higher probability of having an abnormally low DSST score in spring (OR 1.57 95% CI (1.14, 2.17), and a lower chance of having an abnormally low category fluency score in winter (OR 0.74 95% CI (0.58, 0.94) (Fig 2).

Mean and maximum daily temperatures or humidex increased the risk of abnormally low category fluency scores (ORs for 10 degrees, 1.15 (1.01,1.31) and 1.14 (1.02,1.28) respectively) whereas minimum temperature or humidex decreased the risk of abnormally low subjective memory performance scores (OR 0.73 95% CI (0.54,0.98)) (Fig 2).

Sensitivity analysis using three- or ten-day lags were consistent with the main analysis (Supplementary S5, S6 Tables). However, the effect on category fluency didn't persist when using an alternative definition of abnormally low scores (Supplementary S7 Table).

## Weather effects on physical and functional outcomes

A seasonal effect was observed for gait speed, SPPB and balance. In winter, compared to autumn, gait speed increased by 0.0254m/s, SPPB by 0.163 points and balance by 0.0731 points. In summer, gait speed decreased by 0.017 m/s. In spring, SPPB decreased by 0.0534 and balance by 0.0731 points (Supplementary S8 Table).

Temperatures and humidex affected all physical parameters, except ADCS-ADL-PI and chair stands. A 10-degree increase in minimum, mean or maximum temperature respectively decreased gait speed by 0.017m/s, 0.016m/s and 0.0136m/s, SPPB by 0.073, 0.078, and 0.072 points, and balance by 0.0437, 0.045 and 0.04 points. Grip strength decreased significantly with maximum temperature only (0.157 kg for 10 degrees) (S5 Table).

Sensitivity analyses were in accordance with temperature effects for all physical and functional tests (Supplementary S9 and S10 Tables).

## Abnormally low physical performance analysis

Among the 6900 visits, the number of abnormally low scores was 2311 (34.1%) for gait speed, 138 (2.02%) for ADCS-ADL-PI, 786 (11.3%) for SPPB, 124 (1.89%) for chair stands, 238 (3.48%) for balance and 140 (2.17%) for grip strength.

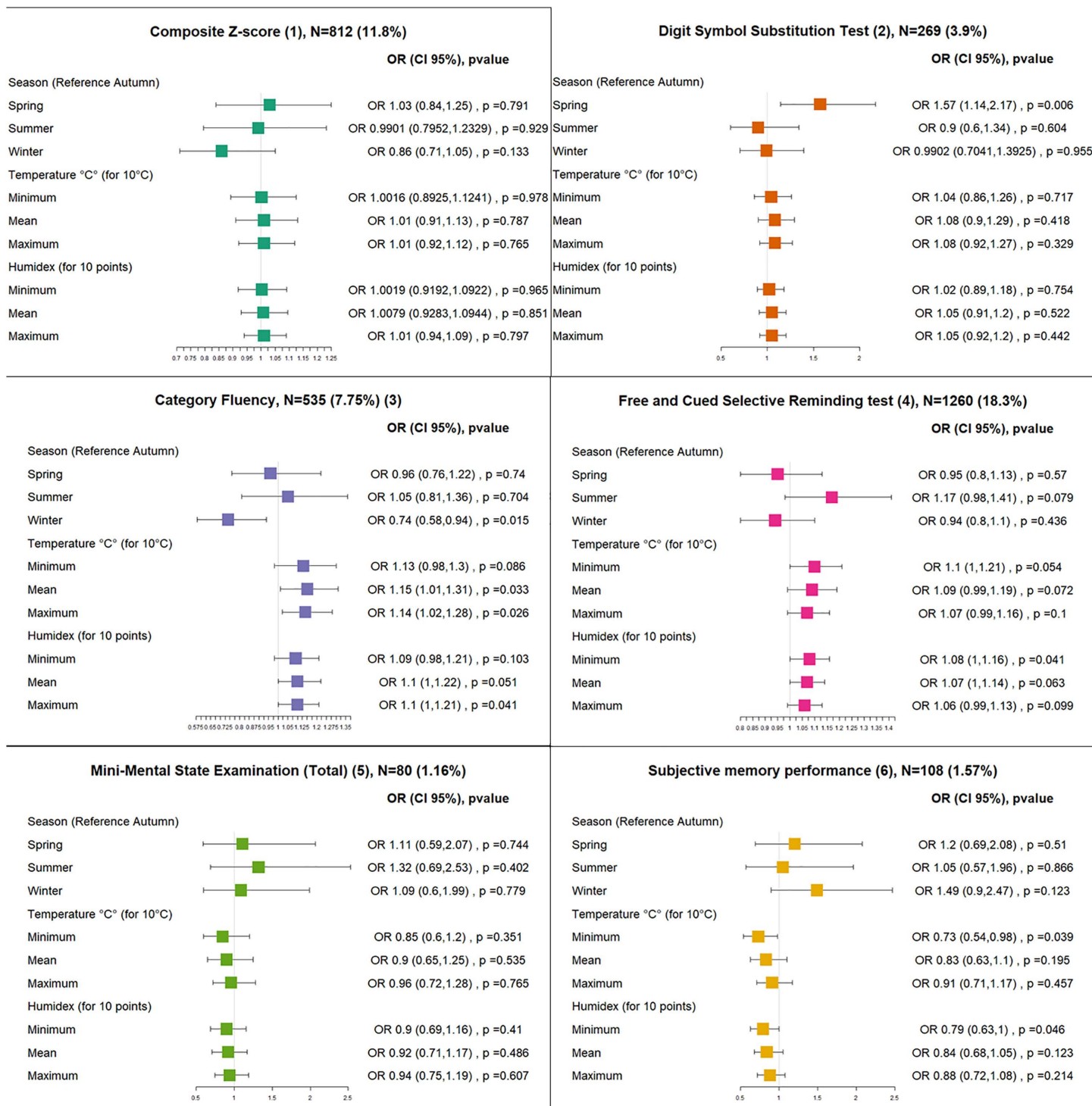

**Fig 2. Effects of the weather on the probability of having an abnormally low score (cognitive outcomes)** OR: Odds ratio, N: number of outliers, the total number of visits is 6900, *p value<0.05, (1) Z score is the mean of the Z scores of Digit Symbol Substitution Test, Category Fluency, Free and Cued Selective Reminding test, Mini-Mental State Examination (Orientation), an observation is considered abnormally low if it's inferior of 0.3 SD (minimal clinically important difference) or more than expected. (2) An observation is considered abnormally low if it's inferior by 6 points (minimal clinically important difference) or more than expected. (3) An observation is considered abnormally low if it's inferior by 5.15 points (minimal detectable change) or

more than expected. (4) Free and total recall, an observation is considered abnormally low if it's inferior by 1.21 points or more than expected. (5) MMSe total, an observation is considered abnormally low if it's inferior by 3 points (minimal clinically important difference) or more than expected. (6) 1 to 100 VAS asking "How well does your memory work?", An observation is considered abnormally low if it's inferior by 24.9 points (minimal detectable change) or more than expected.

Regarding seasonal effects, compared to autumn, there was a higher probability of having an abnormally low gait speed score in summer (OR 1.2 95% CI (1.03, 1.39)), a lower chance of abnormally low SPPB scores in winter and spring (OR 0.76 95% CI (0.62, 0.94)) and spring (OR 0.74 95% CI (0.61, 0.9)), and a higher chance of abnormally low grip strength scores in winter (OR 1.66 (1.02, 2.69) and spring (OR 1.76 95% CI (1.11, 2.81), respectively)) (S6 Table).

Temperature and humidex only affected the probability of an abnormally low gait speed and SPPB scores: 10-degree increases in minimum and mean temperature were associated, respectively, with ORs of 1.13 95% CI (1.04, 1.22) and 1.08 CI 95% (1.001, 1.16) for gait speed, and 1.13 95% CI (1.01, 1.27), 1.15 95% CI (1.03, 1.29) for SPPB. Furthermore, a 10-degree increase in maximum temperature was associated with an OR of 1.16 95% CI (1.05, 1.28) for SPPB (Fig 3).

Data are Median [EIQ], or n (%), unless otherwise specified. (* Higher word counts, speed or score indicates better performance, **Longer time indicates lower performance

Sensitivity analyses were in accordance with temperature and seasonal effects for all physical and functional tests (Supplementary S11 and S12, 13 Tables).

## Discussion

### Summary of main findings

The results from the study conclude that elevated outdoor temperatures were associated with a significantly increased risk of abnormally low gait speed or SPPB, and that temperature was not associated with an increased risk of abnormally low cognitive function. Indeed, maybe due to seasonal depression, subjective memory was actually less likely to be abnormally low when temperatures rose. We did find an effect of the weather on category fluency but it didn't persist in sensitivity analyses.

We showed a small but consistent effect of weather on episodic memory, grip strength, and balance but it didn't lead to abnormally low scores. Season had an effect on some, but not all, cognitive and physical/functional outcomes.

### Comparison with literature

The effect of weather on physical performance reconciles rather well with existing literature. Indeed, Linderman et al., who studied 81 older adults at home in Germany, showed a decrease in gait speed of 0.074m/s per 10-degree increase in inside temperature, whereas we found a decrease of 0.017m/s for 10-degree increases in minimum outside temperature [11]. The result we found is especially important because gait speed is a strong predictor of adverse events in our population and is routinely used to evaluate older adults' mobility [32–34].

In terms of cognitive function, our results partially reconcile with existing literature. Two cross-sectional studies, respectively on 20,687 older adults in the US and 3448 older adults in China also didn't show any significant effect of temperature [10,12]. However, a longitudinal study of 1976 patients followed for up to 10 years in Taiwan showed a significant effect of higher temperature on the probability of having an altered Short Portable Mental Status Questionnaire. Nonetheless, this disparity could be explained by the vastly different weather conditions [9]. Two other studies found results that are more difficult to reconcile with ours. Dai et al showed a U-shaped association of the probability of altered cognition (MMSe < 25) with both extreme cold and extreme heat in a longitudinal study of 594 older US veterans followed for up to eight years [7], and Zhao et al. showed a similar effect (using a composite cognitive Z-score) in a cross-sectional study of

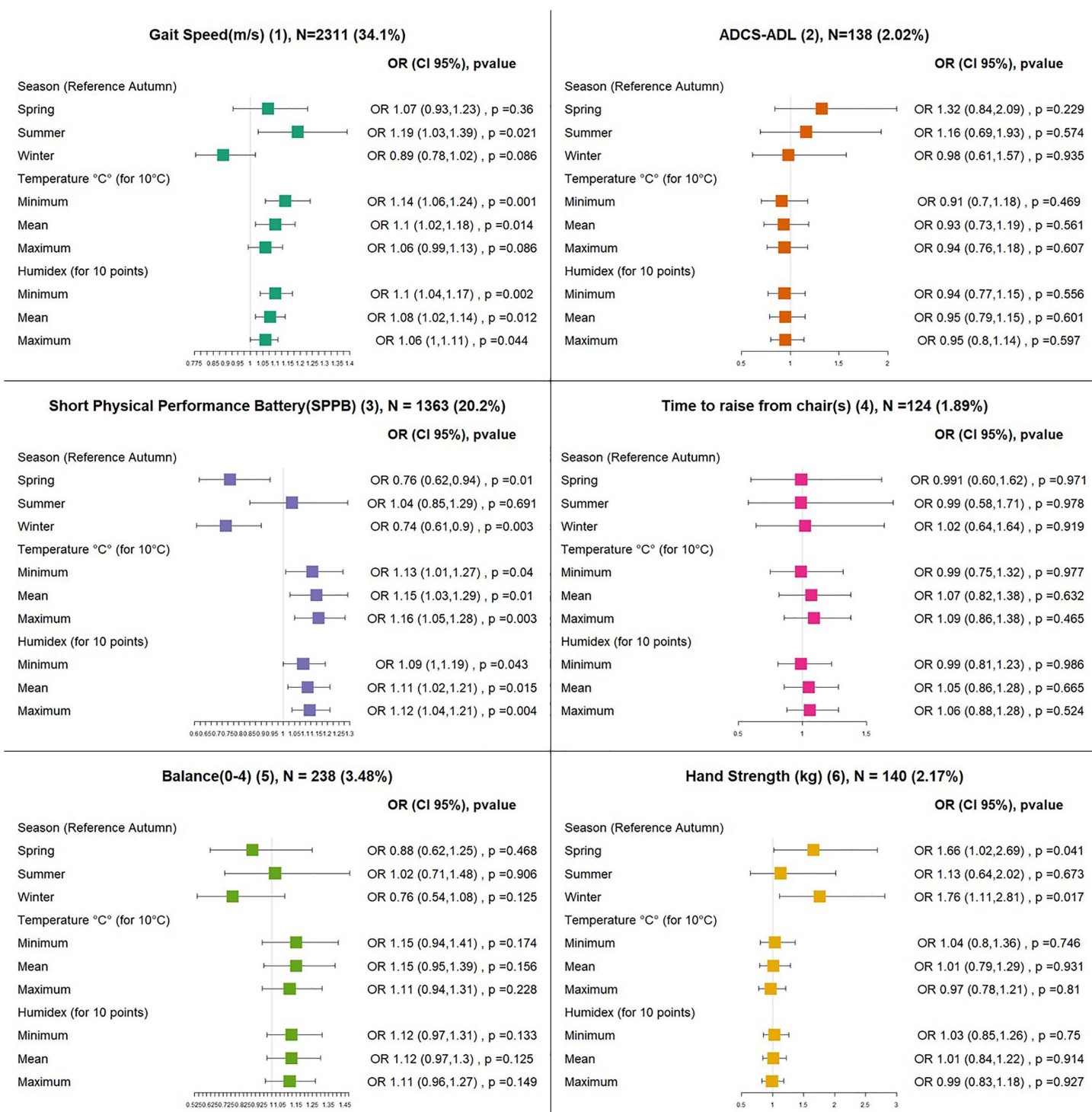

**Fig 3. Effects of the weather on the probability of having an abnormally low score (functional and physical outcomes):** OR: Odds ratio, N: number of outliers, the total number of visits is 6900, *p value<0.05, (1) An observation is considered abnormally low if it's inferior of 0.05 m/s (minimal clinically important difference) or more than expected. (2) An observation is considered abnormally low if it's inferior of 5.58 points (minimal detectable change) or more than expected. (3) An observation is considered abnormally low if it's inferior by 1 points (minimal clinically important difference) or more than expected. (4) An observation is considered abnormally low if it's superior by 5.1 seconds (minimal clinically important difference) or more than expected. (5) An observation is considered abnormally low if it's inferior by 0.63 points (minimal detectable change) or more than expected. An observation is considered abnormally low if it's inferior by 6.5 kg (minimal clinically important difference) or more than expected*p value<0.05.

777 older women in Germany [8]. We had a similar range of mean temperatures (−9.8 to 29.3 in our study versus −5.8 to 25.7 in the study reported by Dai and −6.7 to 29 in the study reported by Zhao) and participant ages (median 75 years in our study versus mean ages of 73 in the studies reported by Dai [7] and 73.5 in the study reported by Zhao [8]). However, Dai et al. failed to exclude patients presenting depressive symptoms [7]. Furthermore, they used a very low MMSe threshold (<25), likely to be more indicative of dementia onset or mild cognitive impairment than of weather-related performance alterations, making it less pertinent in a prevention setting. These two diagnoses have been shown to be related to season or weather conditions and could therefore act as confounders in their study [25,35].

Zhao et al. did exclude participants with depression or dementia, but the association they found might have been overestimated by their use of a reference population to standardise their cognitive score [8]. It might therefore be possible that a variation of one standard deviation in their population is a smaller cognitive variation than in our analysis, even more so considering they had a higher proportion of participants with a low or medium level of education (80.8%) compared to our study (95.5% and 82.5% of patient with medium or high education).

## Methodological strengths and limitations

Our study has some strengths, including the high number of participants and long follow-up that allowed us to model individual cognitive and physical trajectories. We also had a high level of standardisation of cognitive and functional measures as these data came from a randomised clinical trial, as well as a wide array of temperatures in a large geographic area in France.

The main limitation of our study is the granularity of our meteorological observations. Indeed, we used local meteorological stations rather than weather conditions modelled at the residential addresses of participants. However, participants would have to go to their evaluation centre on the day of evaluation and would therefore be exposed to the weather there at the moment of their evaluation. Moreover, MAPT included participants in local centres and we only included centres close to a meteorological station; therefore, the probability that a participant had very different weather at home to that which they were attributed in our analysis is low. A second limitation is our use of simple aggregation techniques for the day's weather (mean, minimum and maximum) or our lags, rather than more complex models [8] and our exclusive use of external weather rather than precise modelling of each environment encountered by participants (including home and transportation). Although this is clearly a limitation in terms of the explanatory value of our models, it facilitates applicability. Indeed, simple aggregation techniques and observed meteorological data could be reasonably employed in clinical and research settings. On the other hand, it could prove very difficult to model patient's entire environment including indoor, transport and facility temperature and humidity. Furthermore, it is important to note that this limitation is conservative, meaning that although we might not show some associations the ones present are unlikely to be inexistent. Third, our definition of abnormally low is dependent on MCID definitions. Indeed, MCIDs in cognition have recently been the object of controversy, and this could explain why we observed a widely different proportion of abnormally low observations among the different scores [36]. However, we performed sensitivity analysis using different thresholds and found robust results for gait speed and SPPB. Finally, our study was limited in its scope. Namely, we didn't explore mechanisms and mediating factors such as mood, hydration, or transportation burden through which weather affects cognitive and physical function, and further studies are needed that could explain why one seems to be affected but not the other.

## Interpretation of differential effects

Indeed, the different effects we found on different functions could seem surprising, but become more understandable once the multiple physiological pathways involved are accounted for. Weather doesn't only affect function directly. To make a comparison, deaths related to heat waves are much more often caused by decompensation of pre-existing diseases than directly caused by heat stress. Even amongst deaths directly related to heat, death due to dehydration is twice as frequent as death due to heat strokes. [37] Therefore, multiple reasons might explain this differential effect. First, cognition might

recover faster than physical function in the examination room. Also, participants may employ physical function to get to the examination centre itself and therefore may be more subject to tiredness. Additionally, mood, on which weather has a very person-dependent effect, might have a mitigating mediating effect on cognition [38]. Adaptive behaviour could also play a role, with participants being more alarmed by cognitive effects rather than physical effects.

### Implications

Correct assessment of physical capacities, including gait speed is of prime importance in clinical trials or geriatric evaluations. In clinical trials, gait speed can be used as an inclusion criterion or as an outcome. In the context of inclusion criteria, not accounting for weather could therefore bar some participants from receiving the intervention and introduce selection bias. Regarding gait speed as an outcome, ignoring weather could lead to measurement bias and loss of power. In the context of routine geriatric evaluations, underevaluating patients' capacity could bar them from receiving important treatments like invasive surgery or chemotherapy.

### Conclusion

Our results suggest that weather should be considered when assessing gait speed or physical function, and evaluations might need to be rescheduled if the weather is too hot. Further studies are needed to determine the exact range of acceptable temperature or humidex for gait speed and other physical function evaluations, and to confirm our absence of associations with cognition. The IPCC (Intergovernmental Panel on Climate Change) indicates that mean temperatures could rise by 1.5°C by 2040 [39]. This could make heat waves up to 12 times more frequent [40]. Due to the increased number of heatwaves, another approach would be to adapt norms and interpretation of decline (Minimal clinical Important difference) depending on the weather. Additionally, the influence of weather on older adults should be accounted for in the design of medical facilities and city planning.

### Supporting information

**S1 Table. Comparison of the weather per treatment arm in MAPT.**
(DOCX)

**S2 Table. Effect of the weather on cognitive outcomes.**
(DOCX)

**S3 Table. Effect of the weather (with 3 days lag) on cognitive outcomes. *p value<0.05.**
(DOCX)

**S4 Table. Effect of the weather (with 10 days lag) on cognitive outcomes. *p value<0.05.**
(DOCX)

**S5 Table. Effects of the weather (With a 3 days lag) on the probability to have an abnormally low scores (cognitive outcomes).**
(DOCX)

**S6 Table. Effects of the weather (With a 10 days lag) on the probability to have an abnormally low scores (cognitive outcomes).**
(DOCX)

**S7 Table. Effects of the weather on the probability to have an abnormally low scores (cognitive outcomes) using an alternative definition of abnormally low.**
(DOCX)

**S8 Table. Effect of the weather on physical and functional outcomes. *p value<0.05.**
(DOCX)

**S9 Table. Effect of the weather ((with 10 days lag) on physical and functional outcomes.**
(DOCX)

**S10 Table. Effects of the weather (with a three days lag) on the probability to have an abnormally low scores (functional and physical outcomes).**
(DOCX)

**S11 Table. Effects of the weather (with a 10 days lag) on the probability to have an abnormally low scores (functional and physical outcomes).**
(DOCX)

**S12 Table. Effects of the weather on the probability to have an abnormally low scores (functional and physical outcomes) using an alternative definition of abnormally.**
(DOCX)

## Acknowledgments

The authors would like to thank all the participants in the MAPT study.

## MAPT Study Group

The members of the MAPT/DSA group are:

Principal investigator: Bruno Vellas (Toulouse); Coordination: Sophie Guyonnet; Project leader: Isabelle Carrié; CRA: Lauréane Brigitte; Investigators: Catherine Faisant, Françoise Lala, Julien Delrieu, Hélène Villars; Psychologists: Emeline Combrouze, Carole Badufle, Audrey Zueras; Methodology, statistical analysis and data management: Sandrine Andrieu, Christelle Cantet, Christophe Morin; Multidomain group: Gabor Abellan Van Kan, Charlotte Dupuy, Yves Rolland (physical and nutritional components), Céline Caillaud, Pierre-Jean Ousset (cognitive component), Françoise Lala (preventive consultation). The cognitive component was designed in collaboration with Sherry Willis from the University of Seattle, and Sylvie Belleville, Brigitte Gilbert and Francine Fontaine from the University of Montreal.

Co-Investigators in associated centres: Jean-François Dartigues, Isabelle Marcet, Fleur Delva, Alexandra Foubert, Sandrine Cerda (Bordeaux); Marie-Noëlle-Cuffi, Corinne Costes (Castres); Olivier Rouaud, Patrick Manckoundia, Valérie Quipourt, Sophie Marilier, Evelyne Franon (Dijon); Lawrence Bories, Marie-Laure Pader, Marie-France Basset, Bruno Lapoujade, Valérie Faure, Michael Li Yung Tong, Christine Malick-Loiseau, Evelyne Cazaban-Campistron (Foix); Françoise Desclaux, Colette Blatge (Lavaur); Thierry Dantoine, Cécile Laubarie-Mouret, Isabelle Saulnier, Jean-Pierre Clément, Marie-Agnès Picat, Laurence Bernard-Bourzeix, Stéphanie Willebois, Iléana Désormais, Noëlle Cardinaud (Limoges); Marc Bonnefoy, Pierre Livet, Pascale Rebaudet, Claire Gédéon, Catherine Burdet, Flavien Terracol (Lyon), Alain Pesce, Stéphanie Roth, Sylvie Chaillou, Sandrine Louchart (Monaco); Kristel Sudres, Nicolas Lebrun, Nadège Barro-Belaygues (Montauban); Jacques Touchon, Karim Bennys, Audrey Gabelle, Aurélia Romano, Lynda Touati, Cécilia Marelli, Cécile Pays (Montpellier); Philippe Robert, Franck Le Duff, Claire Gervais, Sébastien Gonfrier (Nice); Yannick Gasnier and Serge Bordes, Danièle Begorre, Christian Carpuat, Khaled Khales, Jean-François Lefebvre, Samira Misbah El Idrissi, Pierre Skolil, Jean-Pierre Salles (Tarbes).

MRI group: Carole Dufouil (Bordeaux), Stéphane Lehéricy, Marie Chupin, Jean-François Mangin, Ali Bouhayia (Paris); Michèle Allard (Bordeaux); Frédéric Ricolfi (Dijon); Dominique Dubois (Foix); Marie Paule Bonceour Martel (Limoges); François Cotton (Lyon); Alain Bonafé (Montpellier); Stéphane Chanalet (Nice); Françoise Hugon (Tarbes); Fabrice Bonneville, Christophe Cognard, François Chollet (Toulouse).

PET scans group: Pierre Payoux, Thierry Voisin, Julien Delrieu, Sophie Peiffer, Anne Hitzel, (Toulouse); Michèle Allard (Bordeaux); Michel Zanca (Montpellier); Jacques Monteil (Limoges); Jacques Darcourt (Nice).

Medico-economics group: Laurent Molinier, Hélène Derumeaux, Nadège Costa (Toulouse).

Biological sample collection: Bertrand Perret, Claire Vinel, Sylvie Caspar-Bauguil (Toulouse).

Safety management: Pascale Olivier-Abbal

## Author contributions

**Conceptualization:** Jason Shourick, Valérie Lauwers-Cances, Bruno Vellas, Nicola Coley, Sandrine Andrieu.

**Data curation:** Jason Shourick.

**Formal analysis:** Jason Shourick.

**Methodology:** Jason Shourick, Nicola Coley, Sandrine Andrieu.

**Supervision:** Nicola Coley, Sandrine Andrieu.

**Validation:** Jason Shourick.

**Visualization:** Jason Shourick, Valérie Lauwers-Cances.

**Writing – original draft:** Jason Shourick.

**Writing – review & editing:** Jason Shourick, Valérie Lauwers-Cances, Bruno Vellas, Nicola Coley, Sandrine Andrieu.

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
