## [Decision Letter · Decision Letter 0]

12 May 2025

Dear Dr. shourick,

Thank you for submitting your manuscript to PLOS ONE. After careful consideration, we feel that it has merit but does not fully meet PLOS ONE’s publication criteria as it currently stands. Therefore, we invite you to submit a revised version of the manuscript that addresses the points raised during the review process.

We look forward to receiving your revised manuscript.

Kind regards,

Assoc. Prof. Phakkharawat Sittiprapaporn, Ph.D.

Academic Editor

PLOS ONE

Journal Requirements:

“The MAPT study was supported by grants from the Gérontopôle of Toulouse, the French Ministry of Health (PHRC 2008, 2009), Pierre Fabre Research Institute (manufacturer of the omega-3 supplement), ExonHit Therapeutics SA, and Avid Radiopharmaceuticals Inc. The promotion of this study was supported by the University Hospital Centre of Toulouse. The data sharing activity was supported by the Association Monegasque pour la Recherche sur la maladie d’Alzheimer (AMPA) and the INSERM-University of Toulouse III UMR 1295 Research Unit (CERPOP). This secondary analysis didn’t receive any specific funding or grant”

4. One of the noted authors is a group or consortium [MAPT/IHU HealthAge Open Science study group#MAPT/DSA Study Group]. In addition to naming the author group, please list the individual authors and affiliations within this group in the acknowledgments section of your manuscript. Please also indicate clearly a lead author for this group along with a contact email address.

6. Please include captions for your Supporting Information files at the end of your manuscript, and update any in-text citations to match accordingly. Please see our Supporting Information guidelines for more information: http://journals.plos.org/plosone/s/supporting-information .

Reviewers' comments:

Reviewer's Responses to Questions

**Comments to the Author**

1. Is the manuscript technically sound, and do the data support the conclusions?

Reviewer #1: No

Reviewer #2: Yes

Reviewer #3: Partly

2. Has the statistical analysis been performed appropriately and rigorously?

Reviewer #1: Yes

Reviewer #2: Yes

Reviewer #3: I Don't Know

3. Have the authors made all data underlying the findings in their manuscript fully available?

Reviewer #1: Yes

Reviewer #2: No

Reviewer #3: No

4. Is the manuscript presented in an intelligible fashion and written in standard English?

Reviewer #1: Yes

Reviewer #2: Yes

Reviewer #3: Yes

Reviewer #1: I am glad to review the manuscript entitled 'How weather affects clinical outcomes in elderly subjects'. Althogh this is an interesting topic, the clinical implication and conclusion are quite limited. In this study, weather might affect the clinical outomes of elderly patients through the changes of temperature, but there are many influence factors. First of all, patients were not alwaya exposed to the outdoor condictions and indoor temperature is controllable. Sencodly, some emotional sensitive patents might be influenced by the wether, but such factors were not able to be calculated in this study.

Reviewer #2: Dear Dr. Sittiprapaporn,

Thank you for the opportunity to review the manuscript entitled “How weather affects clinical outcomes in elderly subjects”.

The paper addresses an important and often overlooked aspect of geriatric assessment by exploring the impact of weather conditions on cognitive and physical performance in older adults. The findings, particularly regarding the influence of temperature on gait speed and physical function, have clear implications for both clinical practice and future research. I have provided comments and suggestions to enhance the clarity, depth, and rigor of the authors discussion and conclusions. I hope these recommendations will strengthen the manuscript and further its contribution to the field.

Thank you again for the opportunity to contribute to the review process.

Title

1. The title is broad and appeal to a wide audience, however, the term “clinical outcomes” is vague, especially since the study focuses specifically on cognitive and physical performance.

2. “Elderly subjects” is outdated and less respectful language; current best practice prefers terms like “older adults”.

Abstract

3. Objective: The sentence is unclear and slightly awkward. "…in relation to the expected abilities…" lacks clarity. Please rephrase it.

4. Setting: Consider integrating "France" naturally.

5. Participants: Good description, but the inclusion criteria feel jammed into the sentence. Consider breaking into two sentences or using parentheses.

6. Outcomes and Methods

7. The definition of "abnormally low scores" is important but buried. Also, the abbreviation "SPPB" is used before being defined. Introduce abbreviations before use, and perhaps rephrase.

8. Results: Generally, well presented with statistics but could be clearer on what outcomes were not significant.

9. The structure of "Specifically, temperature also associated with..." is awkward.

10. Conclusion is reasonable and practical, but could emphasize the implication for clinical trials and practice more strongly.

Introduction

The introduction presents a relevant and timely research question within the context of aging, climate change, and functional health assessment. However, the narrative is occasionally disjointed and would benefit from clearer transitions, refined phrasing, and improved organization of ideas. Some claims are underdeveloped or insufficiently supported.

11. Several sentences are awkwardly phrased or grammatically incorrect (e.g., “the concepts of healthy aging and intrinsic capacities… have shown their relevance…” is vague and unclear).

12. The logic behind the narrative jumps abruptly from methodological challenges in aging research to climate projections without smooth transitions.

13. The paragraph about IPCC projections interrupts the development of the argument about clinical heterogeneity.

14. The concept of “abnormally low performance unreflective of actual function” is introduced late and could be foregrounded earlier.

15. The difference between weather, temperature, and climate needs to be clearly defined early on.

16. The introduction notes conflicting prior findings, but doesn’t explain what the current study does differently or why it fills a unique gap.

17. Citations are densely packed and not always well integrated into the argument (e.g., references 11–16 are dropped at the end of a sentence without context).

Methods:

18. Consider briefly stating the rationale for including individuals with subjective memory complaints, prefrailty, or slow gait to contextualize their relevance to cognition research.

19. The intervention groups are well-described. However, the structure could be cleaner.

Consider bulleting or rephrasing for clarity

20. Good contextual reference to the MAPT-plus follow-up and participant characteristics. Clarify “without normalization” and explain why individual test scores were also analyzed this way.

Outcomes

21. Cognitive Outcomes: Clarify that the Z-scores were standardized based on baseline means/SDs. Also, state the rationale for expressing Z-scores as percentages (multiplied by 100)—was it for easier interpretation?

22. Functional/Physical Outcomes: For clarity, briefly mention the meaning or significance of the SPPB and ADCS-ADL-PI scores, especially for readers unfamiliar with these instruments.

23. Explanatory Variables: Consider adding a sentence on the relevance of weather to cognitive and physical outcomes in aging populations (e.g., thermal stress, reduced activity levels).

24. Descriptive and Inferential Approach:

A clearer explanation can improve the phrase “time, time², time³” (e.g., “modeled time as a cubic polynomial”) for a more general audience.

25. The logic behind using a 30% SD cutoff for the FCSRT is explained; do the other MCIDs also correspond to clinically meaningful changes in older adults? If so, briefly cite this rationale.

26. State why only bivariate models were used for weather-abnormal scores analysis. Would adjusting for baseline characteristics strengthen the findings?

27. Briefly comment on the proportion of missing data for key variables, if available.

28. Not adjusting for multiple comparisons is acceptable for exploratory work, but this should be acknowledged as a limitation, either here or in the Discussion.

29. Improve clarity and readability in complex sentences (e.g., polynomial modeling, random slopes).

30. Justify methodological decisions in more reader-friendly language (e.g., Z-score % scaling, bivariate regressions).

31. Minor rewording for consistency and grammar (e.g., “made accessed” → “were accessed”).

Results

This results section is strong in terms of content and statistical clarity. With some adjustments in language, transitions, and layout, it could be more accessible to a broader audience while retaining its scientific rigor.

32. The section is rich in data but better flow and transitions between parts can improve it.

33. Repetitions like "respectively" and multiple numbers in one sentence can overwhelm the reader.

34. It might help to group related findings more clearly, especially for readers not familiar with all measures.

35. The main findings are present but readability is slightly hindered by stats-heavy phrasing.

36. Sensitivity analyses with 3- and 10-day lags confirmed the main findings: higher temperatures and humidex were associated with poorer FCSRT scores and improved subjective memory (see Supplementary Tables 3 and 4).

37. Seasonality and Abnormally Low Cognitive Performance is well presented but could use minor structural improvements.

38. Physical/Functional Outcomes – Main Effects are well structured but

39. Physical Measures & Temperature include good detail but some stat-heavy phrasing.

40. Temperature & Abnormally Low Physical Scores are technically sound, just tidy phrasing.

Discussion

41. Currently, the discussion is a bit dense and jumps quickly between ideas. Consider reorganizing it into clearer sub-sections or paragraphs:

• Summary of main findings

• Comparison with literature

• Methodological strengths and limitations

• Interpretation of differential effects

• Implications and future directions

42. The discussion suggests that weather might influence assessment outcomes, especially gait speed. Emphasize how this could affect clinical trials, screening programs, or routine geriatric evaluations.

43. Should clinicians adjust thresholds seasonally? Should testing environments be climate-controlled?

44. Address possible confounders: You discuss depressive symptoms and education as potential confounders in other studies. Consider also discussing whether mood, hydration, or transportation burden could confound your own findings, especially on test days.

45. Tone down the certainty in some comparative statements. For example: "the association they found might have been exaggerated..." . Consider rephrasing to soften speculation unless supported by direct evidence.

Conclusion

46. Add a brief nod to public health implications. A sentence could be added about the importance of this knowledge in population aging, heat waves, or urban planning for older adults.

47. Rephrase "evaluations might need to be rescheduled if the weather is too hot,"

Reviewer #3: 1. This manuscript investigates whether outdoor weather might influence physical and cognitive performance of patients tested in an indoor clinical setting. When I read the abstract, I thought the authors were investigating temperature inside the clinic, and I wondered why their clinic has such variable temperatures. It became clear when reading the introduction that weather outside during the course of an entire day might be more important to a patient’s performance than the temperature of the clinic, which they may only experience briefly. This is a novel and interesting idea. The abstract needs to more clearly describe this approach.

2. Abstract: Define unit of measurement for temperature: Celsius.

3. I appreciate that the authors’ are taking a fresh look at archival data to answer a new question.

4. Table 1: Not enough info is given about what the values mean. The legend/footnote should be expanded.

5. Figures 2 and 3 are so blurry I cannot read them.

6. In the Results section “Effect on cognitive outcomes,” Table 3 is referenced, but there is no Table 3.

7. I find the conclusion of this study intriguing, that outdoor weather should be considered even for indoor testing, and rescheduling may be valuable when weather is very hot. This could have an impact on clinical practice. However, I am unable to assess the validity of the conclusions because the figures are not legible, tables are not well described, and there is a missing table.

**Do you want your identity to be public for this peer review?** For information about this choice, including consent withdrawal, please see our Privacy Policy

Reviewer #1: No

Reviewer #2: **Yes: ** Dr. Marjan Hosseini

Reviewer #3: No

---

## [Author Response · Author response to Decision Letter 1]

16 Jul 2025

1. Is the manuscript technically sound, and do the data support the conclusions?

Reviewer #1: No

Reviewer #2: Yes

Reviewer #3: Partly

2. Has the statistical analysis been performed appropriately and rigorously?

Reviewer #1: Yes

Reviewer #2: Yes

Reviewer #3: I Don't Know

3. Have the authors made all data underlying the findings in their manuscript fully available?

Reviewer #1: Yes

Reviewer #2: No

Reviewer #3: No

4. Is the manuscript presented in an intelligible fashion and written in standard English?

Reviewer #1: Yes

Reviewer #2: Yes

Reviewer #3: Yes

5. Review Comments to the Author

Reviewer #1: I am glad to review the manuscript entitled 'How weather affects clinical outcomes in elderly subjects'. Althogh this is an interesting topic, the clinical implication and conclusion are quite limited. In this study, weather might affect the clinical outomes of elderly patients through the changes of temperature, but there are many influence factors. First of all, patients were not alwaya exposed to the outdoor condictions and indoor temperature is controllable. Sencodly, some emotional sensitive patents might be influenced by the wether, but such factors were not able to be calculated in this study.

Indeed our study could not account for factors such as personal sensitivity or individual indoor, or in transport conditions that probably play a role in mediating weather effects on patients and further study could be performed. However, as we mentioned in the discussion section of the article “Although this is clearly a limitation in terms of the explanatory value of our models, it facilitates applicability. Indeed, simple aggregation techniques and observed meteorological data could be reasonably employed in clinical and research settings.”

Furthermore, this is a rather conservative limitation, meaning that although we might not show some association due to adaptive strategies such indoor climatisation, the associations that we showed are very unlikely to be caused by these adaptive strategies.

Therefore we updated the discussion paragraph (sentences underlined are new) in :”A second limitation is our use of simple aggregation techniques for the day’s weather (mean, minimum and maximum) or our lags, rather than more complex models12 and our exclusive use of external weather rather than precise modeling of each environment encountered by participants (including home and transportation) . Although this is clearly a limitation in terms of the explanatory value of our models, it facilitates applicability. Indeed, simple aggregation techniques and observed meteorological data could be reasonably employed in clinical and research settings. On the other hand, it could prove very difficult to model participants’ entire environment including indoor, transport and facility temperature and humidity. Furthermore, it is important to note that this limitation is conservative, meaning that although we might not show some associations the ones present are unlikely to be false positive. “

Reviewer #2: Dear Dr. Sittiprapaporn,

Thank you for the opportunity to review the manuscript entitled “How weather affects clinical outcomes in elderly subjects”.

The paper addresses an important and often overlooked aspect of geriatric assessment by exploring the impact of weather conditions on cognitive and physical performance in older adults. The findings, particularly regarding the influence of temperature on gait speed and physical function, have clear implications for both clinical practice and future research. I have provided comments and suggestions to enhance the clarity, depth, and rigor of the authors discussion and conclusions. I hope these recommendations will strengthen the manuscript and further its contribution to the field.

Thank you again for the opportunity to contribute to the review process.

Title

1. The title is broad and appeal to a wide audience, however, the term “clinical outcomes” is vague, especially since the study focuses specifically on cognitive and physical performance.

We changed the title for “How weather affects cognitive and physical outcomes in older adults”

2. “Elderly subjects” is outdated and less respectful language; current best practice prefers terms like “older adults”.

We changed the title for “How weather affects cognitive and physical outcomes in older adults”

Abstract

3. Objective: The sentence is unclear and slightly awkward. "…in relation to the expected abilities…" lacks clarity. Please rephrase it.

We changed the objective to” “To ascertain whether, in comparison to the participants' expected abilities, the weather may cause abnormally poor cognitive or physical performance.”

4. Setting: Consider integrating "France" naturally.

We changed the setting to “Study conducted between May, 2008, and Feb, 2011 in 13 memory centres in France and Monaco”

5. Participants: Good description, but the inclusion criteria feel jammed into the sentence. Consider breaking into two sentences or using parentheses.

We changed the participants section in : “1313 participants from the MAPT trial, a 5-year multicentre prevention trial, which included dementia-free individuals aged over 70 years. Participants presented subjective memory complaints, slow gait speed and/or an IADL limitation.”

6. Outcomes and Methods

7. The definition of "abnormally low scores" is important but buried. Also, the abbreviation "SPPB" is used before being defined. Introduce abbreviations before use, and perhaps rephrase.

The SPPB abbreviation was introduced in the main outcome section at the time of its first use (line 23 p7). We modified the abnormally low score definition as such: “Abnormally low scores were defined as an observed score that was lower than the individual's expected ability by at least the minimal clinical important difference.”

8. Results: Generally, well presented with statistics but could be clearer on what outcomes were not significant.

We changed the result section to: “Higher temperature was associated with a significantly increased risk of abnormally low gait speed or SPPB (respectively OR 1.13 95% CI (1.04, 1.22) and OR 1.15 95% CI (1.03, 1.29) for 10 degrees), but did not significantly increase the risk of abnormally low cognitive function”.

9. The structure of "Specifically, temperature also associated with..." is awkward.

This sentence was deleted.

10. Conclusion is reasonable and practical, but could emphasize the implication for clinical trials and practice more strongly.

We modified the conclusion to: “Our results suggest that weather conditions should be strongly considered when assessing the physical performance of older adults in the context of clinical practice and clinical research, as examination in hot weather might lead to false conclusions on the participants' abilities.”

Introduction

The introduction presents a relevant and timely research question within the context of aging, climate change, and functional health assessment. However, the narrative is occasionally disjointed and would benefit from clearer transitions, refined phrasing, and improved organization of ideas. Some claims are underdeveloped or insufficiently supported.

11. Several sentences are awkwardly phrased or grammatically incorrect (e.g., “the concepts of healthy aging and intrinsic capacities… have shown their relevance…” is vague and unclear).

This sentence has been rephrased as “The concepts of healthy aging and intrinsic capacities—which place more emphasis on function than diseases—have proven useful in forecasting the health, mortality, and quality of life of older persons in the setting of a rapidly aging population.”

12. The logic behind the narrative jumps abruptly from methodological challenges in aging research to climate projections without smooth transitions.

We deleted all mention of climate projection from the introduction.

13. The paragraph about IPCC projections interrupts the development of the argument about clinical heterogeneity.

See answer above, additionally weather projection and effects are necessary context to explain how and why they might play a role in heterogeneity.

14. The concept of “abnormally low performance unreflective of actual function” is introduced late and could be foregrounded earlier.

The rest of the elements are necessary context to introduce this notion, which, without the developments on heterogeneity and weather as a cause of heterogeneity would be difficult to comprehend.

15. The difference between weather, temperature, and climate needs to be clearly defined early on.

We added the following sentence : “However, weather, i.e. the state of the atmosphere at a given time and place, affects individuals throughout the day and in more complex ways than temperature during evaluation.”

16. The introduction notes conflicting prior findings, but doesn’t explain what the current study does differently or why it fills a unique gap.

We gave the following justification in the introduction : Furthermore, these studies did not try to determine whether the performance of an individual could be unreflective of their actual function because of the weather at the time it was evaluated.

17. Citations are densely packed and not always well integrated into the argument (e.g., references 11–16 are dropped at the end of a sentence without context).

In line with standard practice, we kept our introduction relatively brief and did not present these studies at great length. However, there is a more detailed discussion of their results in the discussion section.

Methods:

18. Consider briefly stating the rationale for including individuals with subjective memory complaints, prefrailty, or slow gait to contextualize their relevance to cognition research.

This study reuses the data from the MAPT trial. The rationale for the inclusion criteria is explained in Gillette-Guyonnet S et al. Alzheimers Dement. 2009;5(2):114-21, PMID: 19328438. Briefly, the rationale was that frailty was considered to be a risk factor for cognitive decline and dementia in older people. At the time of the study, there was no consensus about the definition of frailty, and so it was decided to use three criteria thought to reflect cognitive and/or physical frailty, and to target individuals towards the “pre-frail” end of the frailty spectrum.

We have added the following sentence to the methods section to clarify this point: "These criteria were chosen to identify people who are believed to be at a higher risk of cognitive decline or dementia and who are in the early stages of cognitive and/or physical frailty." and the following reference : Gillette-Guyonnet S et al. Alzheimers Dement. 2009;5(2):114-21, PMID: 19328438

19. The intervention groups are well-described. However, the structure could be cleaner.

Consider bulleting or rephrasing for clarity

We bulleted the intervention groups

20. Good contextual reference to the MAPT-plus follow-up and participant characteristics. Clarify “without normalization” and explain why individual test scores were also analyzed this way.

We clarified as follows: “without normalization i.e Z-score transformation, as this is the way they would be used in clinical practice”

We added the following sentence :

Outcomes

21. Cognitive Outcomes: Clarify that the Z-scores were standardized based on baseline means/SDs.

We added the following sentence : “As in the MAPT trial, the Z score was calculated using baseline means and standard deviation.”

Also, state the rationale for expressing Z-scores as percentages (multiplied by 100)—was it for easier interpretation?

Indeed, please refer to the sentence “To facilitate readability, the composite Z-score is presented as a percentage (i.e. multiplied by 100).”

22. Functional/Physical Outcomes: For clarity, briefly mention the meaning or significance of the SPPB and ADCS-ADL-PI scores, especially for readers unfamiliar with these instruments.

We added the following explanations : The SPPB is a global measure of physical capacity, and mobility which comprises three evaluations (Gait Speed, Time to rise from chair and balance) each scored on a 4 point scale. The ADCS-ADL-PI is a fifteen item questionnaire developed to assess early alterations of activities of daily living (like preparing meals or getting dressed).

23. Explanatory Variables: Consider adding a sentence on the relevance of weather to cognitive and physical outcomes in aging populations (e.g., thermal stress, reduced activity levels).

The aim of this article is to evaluate the link between weather and these performances. The justification for performing this study therefore lies mainly in the introduction with sentences such as : “This could make heat waves up to 12 times more frequent. These extreme events have a well-known effect on morbidity and mortality, especially amongst older adults”

24. Descriptive and Inferential Approach:

A clearer explanation can improve the phrase “time, time², time³” (e.g., “modeled time as a cubic polynomial”) for a more general audience.

We modified the sentence as suggested: “Initial models included: participant-specific random slopes and intercepts, the fixed effects of time modeled time as a cubic polynomial, age and its interaction with time polynomials, sex, level of education and centre”

25. The logic behind using a 30% SD cutoff for the FCSRT is explained; do the other MCIDs also correspond to clinically meaningful changes in older adults? If so, briefly cite this rationale.

We added the following sentence : Indeed the minimal important clinical difference (MCID represents clinical meaningful change, which could therefore lead to clinical consequences (decision not to treat).

26. State why only bivariate models were used for weather-abnormal scores analysis. Would adjusting for baseline characteristics strengthen the findings?

We thank the reviewer for this remark. H however, abnormally scores are defined from residuals of the model including baseline characteristics. In linear mixed models models, residuals are independent from explanatory variables. For more clarity we added the following sentence : “Because residuals are independent from explanatory variables we did not include age, sex or education level in the abnormally low score analysis.”

27. Briefly comment on the proportion of missing data for key variables, if available.

As mentioned in the flowchart, 31 patients had a missing value for level of education. As mentioned in line 11 p 10, using a linear mixed model, means there are no other missing values for participants as slopes are modelled on all available visits using a continuous time scale. There was no day without any meteorological data.

28. Not adjusting for multiple comparisons is acceptable for exploratory work,

---

## [Decision Letter · Decision Letter 1]

18 Sep 2025

Dear Dr. shourick,

Thank you for submitting your manuscript to PLOS ONE. After careful consideration, we feel that it has merit but does not fully meet PLOS ONE’s publication criteria as it currently stands. Therefore, we invite you to submit a revised version of the manuscript that addresses the points raised during the review process.

We look forward to receiving your revised manuscript.

Kind regards,

Assoc. Prof. Phakkharawat Sittiprapaporn, Ph.D.

Academic Editor

PLOS ONE

Journal Requirements:

Reviewer's Responses to Questions

**Comments to the Author**

Reviewer #4: All comments have been addressed

Reviewer #5: All comments have been addressed

2. Is the manuscript technically sound, and do the data support the conclusions?

Reviewer #4: Yes

Reviewer #5: Yes

3. Has the statistical analysis been performed appropriately and rigorously?

Reviewer #4: Yes

Reviewer #5: Yes

4. Have the authors made all data underlying the findings in their manuscript fully available?

Reviewer #4: Yes

Reviewer #5: Yes

5. Is the manuscript presented in an intelligible fashion and written in standard English?

Reviewer #4: Yes

Reviewer #5: Yes

Reviewer #4: This is a technically sound paper, the data support the conclusions, and the revisions adequately address most of the reviewers' comments. The conclusion that outdoor weather is associated with impaired physical performance in clinical settings is important and worthy of publication. One minor point: the introduction shows signs of having been written with the help of artificial intelligence, as long dashes in the sentence are usually used by AI, and the highlighting of individual words is also used by AI.

Reviewer #5: I am honored for the opportunity to review your manuscript entitled “How weather affects clinical outcomes in elderly subjects.” This study addressed a unique perspective better understanding the health of elderly adults and how it is impacted by the weather. I have listed a few recommendations that address grammatical and punctuation errors as well as some suggestions to strengthen the clarity of the manuscript.

Introduction

Consider rephrasing ‘older persons’ to ‘elderly adults’

Consider removing ‘Indeed’ and going straight into ‘Targeting function…’ or ‘There are challenges faced when targeting function rather than…’

Consider changing “Because of’ to “Due to…”

The sentence on line 21 is a bit odd, consider changing “was” to “with” and cutting out repeated words (mostly, studies/studying).

Add a space after i.e (line 23).

Discussion

Exclude “Our study has two main results.” and replace it with “The results from the study conclude that…”

Remove extra space on line 7.

**Do you want your identity to be public for this peer review?** For information about this choice, including consent withdrawal, please see our Privacy Policy

Reviewer #4: No

Reviewer #5: No

---

## [Author Response · Author response to Decision Letter 2]

26 Sep 2025

We thank the reviewers for their sound comments.

Reviewer #4: This is a technically sound paper, the data support the conclusions, and the revisions adequately address most of the reviewers' comments. The conclusion that outdoor weather is associated with impaired physical performance in clinical settings is important and worthy of publication. One minor point: the introduction shows signs of having been written with the help of artificial intelligence, as long dashes in the sentence are usually used by AI, and the highlighting of individual words is also used by AI.

We thank the reviewer for this verification, but in fact we did not use any AI tools during the writing of the manuscript. We used a commonly available AI detection tool to try and change the aforementioned signs but unsurprisingly given that no AI was used, no AI was detected. Furthermore, to avoid confusion we deleted highlighting and long dashes.

Reviewer #5: I am honored for the opportunity to review your manuscript entitled “How weather affects clinical outcomes in elderly subjects.” This study addressed a unique perspective better understanding the health of elderly adults and how it is impacted by the weather. I have listed a few recommendations that address grammatical and punctuation errors as well as some suggestions to strengthen the clarity of the manuscript.

We thank the reviewer and have addressed all the suggested changes.

Introduction

Consider rephrasing ‘older persons’ to ‘elderly adults’ :

We have rephrased to “older adults”. We used “older” rather than “elderly” in light of the previous reviewer’s comments.

Consider removing ‘Indeed’ and going straight into ‘Targeting function…’ or ‘There are challenges faced when targeting function rather than…

We made the suggested change.

Consider changing “Because of’ to “Due to…”

We made the suggested change.

The sentence on line 21 is a bit odd, consider changing “was” to “with” and cutting out repeated words (mostly, studies/studying).

We changed the sentence to: “Furthermore, most previous work consisted of physiological studies of small groups of healthy young adults in highly controlled experiments, rather than everyday life”

Add a space after i.e (line 23).

We made the suggested change.

Discussion

Exclude “Our study has two main results.” and replace it with “The results from the study conclude that…”

We made the suggested change.

Remove extra space on line 7.

We made the suggested change.

---

## [Editor Report · Decision Letter 2]

16 Oct 2025

How weather affects cognitive and physical outcomes in older adults

PONE-D-25-14640R2

Dear Dr. shourick,

We’re pleased to inform you that your manuscript has been judged scientifically suitable for publication and will be formally accepted for publication once it meets all outstanding technical requirements.

Kind regards,

Assoc. Prof. Phakkharawat Sittiprapaporn, Ph.D.

Academic Editor

PLOS ONE

---

## [Editor Report · Acceptance letter]

PONE-D-25-14640R2

PLOS ONE

Dear Dr. shourick,

I'm pleased to inform you that your manuscript has been deemed suitable for publication in PLOS ONE. Congratulations! Your manuscript is now being handed over to our production team.

Kind regards,

on behalf of

Assoc. Prof. Dr. Phakkharawat Sittiprapaporn

Academic Editor

PLOS ONE